# Trends in Antidepressants Use in Spain between 2015 and 2018: Analyses from a Population-Based Registry Study with Reference to Driving

**DOI:** 10.3390/ph13040061

**Published:** 2020-04-03

**Authors:** Eduardo Gutiérrez-Abejón, Francisco Herrera-Gómez, Paloma Criado-Espegel, F. Javier Álvarez

**Affiliations:** 1Pharmacological Big Data Laboratory, Pharmacology and Therapeutics, Faculty of Medicine, University of Valladolid, 47005 Valladolid, Spain, alvarez@med.uva.es (F.J.Á.); 2Technical Direction of Pharmaceutical Assistance, Gerencia Regional de Salud de Castilla y León, 47007 Valladolid, Spain; 3Nephrology, Hospital Virgen de la Concha—Sanidad de Castilla y León, 49022 Zamora, Spain; 4Gerencia de Asistencia Sanitaria-Sanidad de Castilla y León, 34001 Palencia, Spain; pcriado@saludcastillayleon.es; 5CEIm, Hospital Clínico Universitario de Valladolid—Sanidad de Castilla y León, 47003 Valladolid, Spain

**Keywords:** antidepressants, driving impairing medicines, automobile driving, drug utilization, traffic accidents

## Abstract

Antidepressants are considered driving-impairing medicines (DIM). This is a population-based registry study that shows the trend in the use of antidepressants in Castile and León, Spain, from 2015 to 2018. Data on antidepressant dispensations at pharmacies and the adjusted use of these medicines by the driver population are presented. For the purposes of analysis, population distribution by age and gender has been taken into account, as well as the three Driving Under the Influence of Drugs, alcohol, and medicines (DRUID) categories. Antidepressants were used by 8.56% of the general population and 5.66% of drivers. Antidepressants were used more commonly by females than by males (12.12% vs. 4.87%, χ² = 1325.124, p = 0.001), and users increased as the age increased, even if women who drive used less antidepressants after turning 60 years of age. Chronic use of antidepressants was relevant (8.28%) in the same way as daily use (3.15%). Most of the consumption included SSRIs (4.99%), which are also known as “other antidepressants” (3.71%). Regardless of antidepressants consumed, users took 2.75 ± 1.19 DIMs, which are mainly anxiolytics (58.80%) and opioids (26.43%). Lastly, regarding consumption of antidepressants according to the DRUID classification, category I predominated over categories II and III. Our findings should serve as a starting point for health and traffic authorities to raise awareness of the risk for traffic accidents, especially involving SSRIs.

## 1. Introduction

Depression causes severe disabilities in the occupational and social life of patients [1] and it is currently recognized as a global public health concern [2]. According to the World Health Organization (WHO), major depressive disorder will become the second leading cause of disability around the world in 2030 [2]. Depression affects nearly 10% of the adult population, and has a lifetime prevalence of approximately 13% [3]. In addition, depression is the ninth most frequent chronic disease among people over 14 years of age, which is more frequent in women than in men [4].

Depression is characterized by loss of energy and cognitive impairment of the patient [5], and it is associated with lethargy and sleep disturbances, which affect daily functions [6], including the fitness to drive [7]. Mobility is fundamental for the independence of the individual in society today [8], and different studies indicate that 80% of patients with depression have a valid driver’s license and 70% drive regularly [9]. Although the objective of pharmacological treatment with antidepressants is long-term remission of symptoms and improved daily activity of patients [8], antidepressants can affect the safety of driving [9,10].

Antidepressants that hold the Anatomical Therapeutic Chemical (ATC) code N06 are classified in four subgroups [11]: (1) non-selective monoamine reuptake inhibitors (or tricyclic antidepressants, TCAs) (N06A), (2) selective serotonin reuptake inhibitors (SSRIs) (N06B), (3) non-selective monoamine oxidase A inhibitors (N06AF), (4) monoamine oxidase A inhibitors (N06AG), and (5) “other antidepressants” (N06AX). Each of these antidepressant types has a particular side-effects profile. For instance, TCAs can cause sedation, dizziness, double and blurred vision, tachycardia, and a tremor [12]. SSRIs are associated with nausea, diarrhea, insomnia, nervousness, agitation, and anxiety [13]. Sedation and the impairment of driver’s abilities are important with TCAs (e.g., amitriptyline, nortriptyline) and the noradrenergic and specific serotonergic antidepressants (NaSSAs) (e.g., mianserine, mirtazapine) belonging to the N06AX subgroup. On the contrary, SSRIs (e.g., as paroxetine, fluoxetine, escitalopram), the other serotonin receptor antagonist and reuptake inhibitors (SARIs) (e.g., trazodone), and the serotonin norepinephrine reuptake inhibitors (SNRIs) (e.g., venlafaxine, duloxetine) are well tolerated considering their lower sedative effect.

Currently, SSRIs have practically replaced TCAs due to their greater tolerability and safety [13]. Studies report on Odds Ratios (OR) from 1.10 to 3.10 for the risk of a traffic accident when being under antidepressant treatment [14], which highlights the risk of increasing doses of TCAs [15,16,17,18] as a clear relation involving exclusively the use of TCAs that is not confirmed [19]. On the other hand, among studies that assessed the influence of non-sedative antidepressants, such as SSRIs and venlafaxine, a slight or no increase in the risk of traffic accidents was found [18,19].

Importantly, antidepressants are present in the three Driving Under the Influence of Drugs, alcohol and medicines (DRUID) categories: SSRIs and monoamine oxidase A inhibitors belong to category I (minor influence on fitness to drive), TCAs belong to categories II (moderate influence on fitness to drive), and III (severe influence on fitness to drive), and the so-called “other antidepressants” (ATC subgroup N06AX) may be encountered into category III (e.g., mianserin, trazodone), or categories I and II (e.g., venlafaxine, duloxetine) (Appendix A) [20].

Trends in antidepressants use by the population is a topic of great interest. Since commercialization of SSRIs, an increase in their use for two to three times may be observed in western countries [21,22]. For instance, in Spain, the use of antidepressants increased by 200% between 2000 and 2013, and, in a similar way as in European Union (EU) countries, SSRIs were the most consumed (70.4%) [23]. However, there is still little information regarding the use of SSRIs and driving risks. In Spain, a specific pictogram on “medicines and driving” was introduced in 2011 on the packaging of all driving-impairing medicines (DIM), including all antidepressants. The main objective of this pictogram was to improve patient and physician knowledge about the effects of these medicines on the ability to drive safely [24].

This study presents the data on antidepressants consumption from 2015 to 2018 in Castile and León, Spain, based on dispensation of such DIMs at pharmacies. The adjusted use of antidepressants by the driver population is also presented in order to assess differences in patterns of use with respect to the general population [24,25,26,27]. In addition, the duration and their concomitant use of antidepressants with other driving impairing medicines were also assessed. Distributions by age and gender, and the use of antidepressants into the three DRUID categories were considered [20].

## 2. Experimental Section

This manuscript presents findings from a population-based registry study carried out according to the Reporting of studies Conducted using Observational Routinely-collected Data (RECORD) recommendations in order to adequately provide real-world evidence into the topic addressed [28]. Our findings cover dispensation of all commercialized antidepressants in Castile and León, Spain (ATC subgroups N06AA, N06AB, N06AG, and N06AX) from 2015 to 2018.

For a better understanding of our results, separately, antidepressants into each of DRUID categories were studied (Appendix A). The data on dispensation of antidepressants and other DIMs were accessible from CONCYLIA (http://www.saludcastillayleon.es/portalmedicamento/es/indicadores-informes/concylia), where the entire information on pharmaceuticals care in Castile and León is routinely collected. The drivers’ license census data for the years 2015 to 2018 were also accessible (http://www.dgt.es/es/seguridad-vial/estadisticas-e-indicadores/permisos-conduccion/) to calculate the adjusted use of antidepressants by drivers (Appendix A), as previously made [24,25,26,27].

For the purposes of an analysis in a real-world setting, all dispensations were considered as equivalent to consumption. Although consumption of antidepressants in hospitals and private clinics was not considered, our findings cover more than 95% of uses since the entire Spanish population is included in the public health system and antidepressants are mandatory prescription drugs.

The following variables were considered: (1) yearly frequency of antidepressants consumption, (2) acute (1–7 days), sub-acute (8–29 days), and chronic use (≥30 days) of antidepressants each year, (3) yearly frequency of daily use of antidepressants, (4) yearly frequency of antidepressants consumption by DRUID categories, and (5) concomitant use of antidepressants with other DIMs during each year. All analyses were done considering age and gender distributions. This study was approved by our local ethics committee on 17 March 2016 (reference number PI 16-387).

Results are expressed as frequencies (percentage) with their corresponding 95% confidence interval (95% CI) or as means accompanied by their standard deviations (SD). Differences between continuous variables were calculated using Student’s t-test, and those between categorical variables were calculated using Pearson’s chi-squared test. The Cochran-Armitage trend test was used to evaluate the consumption trend by years. The level of significance was set at p ≤ 0.05. All statistical analyses were performed by using the Statistical Package for the Social Sciences (SPSS version 24.0., SPSS Inc, Chicago, IL). Lastly, Microsoft Word and Excel (Microsoft Office version 365, Microsoft, Redmon, WA, USA) were used for preparing this manuscript.

## 3. Results

From 2015 to 2018, 8.56% of the general population took at least one antidepressant. Their use was three times greater in women than in men (12.12% vs. 4.87%, χ² = 1325.124, p = 0.001, Table 1), and, in both sexes, the consumption increased as the age increased (Figure 1), which involved the four ATC subgroups (Figure 2). Overall, yearly antidepressant consumption increased 21.18% (Z = 57.216, p < 0.0001) from 2015 (7.64%) to 2018 (9.26%), and this increase involved particularly chronic use (21.80%, Z = 57.607, p < 0.0001) (7.38% in 2015 vs. 8.98 in 2018) and daily use (25.80%, Z = 43.183, p < 0.0001) (2.76% in 2015 vs 3.48 in 2018). Regarding the increase in the study period, no differences were observed between men and women (Figure 3). Chronic use (≥30 days) was the most frequent (97.73%), but 3.15% of the population took at least one antidepressant daily (Table 1). In addition, concomitant use of antidepressants with other DIMs was common. Regardless of antidepressants consumed (Table 2), yearly users took 2.75 ± 1.19 DIMs (anxiolytics, 58.80%, opioids, 26.43%) and daily users took 2.91 ± 1.21 DIMs (anxiolytics, 67.18%, opioids, 28.89%).

Surprisingly, among the 40 DIMs with the highest number of packages dispensed in Castile and León during the covered period, 11 were antidepressants, especially escitalopram, sertraline, mirtazapine, and venlafaxine (Appendix A). SSRIs were the most consumed type of antidepressants (4.99%) regardless of gender, which is followed by the “other antidepressants” (ATC subgroup N06AX) (3.71%) and TCAs (1.11%) (Table 1). The increase in the use of new N06AX antidepressants should also be highlighted (Appendix A): desvenlafaxine (56.90%), mirtazapine (34.68%), and sertraline (32.44%). Among daily users, 7 out of 10 were taking SSRIs. TCAs appeared only related to acute use (<7 days) that was infrequent (0.05%). The use of monoamine oxidase A inhibitors (N06AG) was negligible (18 patients into the covered period, 8.78*10–6%).

Consumption of antidepressants in the DRUID category I was twice compared to those in category II and 1.5 times higher than those in category III. Nevertheless, the highest increase in the use corresponded to antidepressants in category III (40.84%), which is followed by those in categories II (18.84%) and I (13.44%) (Figure 4 and Appendix A).

The results obtained for drivers were similar: 5.66% used antidepressants and 1.93% took these medicines every day (Table 1). As for the general population, chronic use was the most frequent. The use of antidepressants increased with age, and, in the case of women, a peak was reached in the age range group of 55–59 years, which is 20 years earlier than in men (Figure 1). There were no differences in concomitant DIM use between daily and non-daily users (Table 2).

## 4. Discussion

Our results show that the increase in antidepressants consumption between 2015 and 2018 in Castile and León, Spain was relevant. In each year of the study period, women users predominated over men users in both the general population and the driver population. With respect to the type of antidepressant, SSRIs were the most commonly used. In addition, antidepressants were mostly consumed chronically, and acute and subacute use was practically insignificant. More than half of individuals who used antidepressants during the study period also used other DIMs concomitantly, which can seriously influence fitness to drive.

The increase in the prevalence of antidepressant use is consistent with other national data and comparable with those obtained in other European countries: SSRIs and the “other antidepressants” (ATC subgroup N06AX) were the most consumed [21,22,23]. It is important to differentiate between classic N06AX antidepressants, such as mianserin or trazodone, that are less used, and the most recent ones, such as venlafaxine, mirtazapine, etc., which are trendy.

Antidepressants are the second most prescribed psychotropic medication for the elderly, with the first being benzodiazepines [29]. According to our results, consumption in patients in the age range group of 70–74 years was significantly higher than in younger individuals, and this fact is consistent with the results emanating from other studies that present a prevalence of 11% in North America [30] and 12% in Europe (France) [31]. Older women used the most common antidepressants. Likely, high predisposition for depressive disorders among older women may explain the use of antidepressants observed [32].

Notwithstanding, the increase in antidepressants consumption may be due to other factors different from the prevalence of depression itself. An increase in detection by physicians, and an increase in clinical indications for antidepressants, e.g., neuropathic pain (TCAs) or for smoking cessation (bupropion) [23], should be taken into account. New clinical indications have appeared because different effects on cognitive function require antidepressants at doses substantially lower than those for depression [29]. A relevant percentage of antidepressant prescriptions, i.e., amitriptyline and duloxetine, may be due to chronic pain (prevalence of 62% in people over 75 years of age) [21].

Antidepressants are mainly used chronically (≥30 days). A total of 40% of antidepressants were prescribed for more than 180 days, which is, according to the average use of these medications, 251 to 525 days for SSRIs that are frequently switched to SNRIs [21]. Furthermore, antidepressants are not used as monotherapy. Particularly in the elderly, the concomitant use of benzodiazepines to treat anxiety is frequent. Traffic accidents increase when using both antidepressants and benzodiazepines including 37% in the case of SSRIs and 54% in the case of TCAs [33].

The effect on fitness to drive is not only due to medication. The depressive disorder also influences this factor [34]. Depression is associated with slower reaction time in the driving simulator [35], and patients with these disorders have deficits in cognitive abilities that can cause driving errors and intensify aggressive responses to frustrating roadway events [34]. Additionally, depression is often associated with suicidal ideation, which may increase the risk of an unintended and self-harm traffic accident [3].

Remarkably, the evidence on how antidepressants affect fitness to drive is contradictory, and there are studies that did not support a direct relation [36]. For example, in the case of mirtazapine, there are studies that indicate an important driving impairment on the first day of administration, which decreases thereafter [37]. Similar observations may be found in the case of TCAs, where gradual tolerance may be affirmed [38]. Particularly, experimental studies highlight the significance of the deterioration of driving abilities when compared to healthy controls [5], but these findings are pending to confirm in large real-world prospective assessments [3]. Furthermore, controversies when classifying SSRIs by different existing classifications should be considered. The French Agency for Health Products Safety considers these antidepressants into level 2, i.e., these medicines could affect the ability to drive and require medical advice from a physician or a pharmacist before use. According to the International Council on Alcohol, Drugs, and Traffic Safety (ICADTS), SSRIs belong to category I, i.e., these medicines are presumed to be well tolerated or unlikely to produce an effect on driving abilities [10]. Lastly, according to the DRUID classification, SSRIs are included in category I, which has a minor influence on fitness to drive [20]. In general terms, SSRIs are being considered to have a lower impact on fitness to drive, even though OR > 2 may be found [39,40].

On the other hand, the conclusions obtained by Ravera et al. [10] and Brunnauer et al. [8,14] when analyzing different experimental studies indicate that SSRIs do not present a high risk for driving vehicles, unless treatment is at high doses or there is concomitant consumption with other DIMs.

It is considered important to remember that the psychomotor impairment of the patient with depression may have a greater impact on the increased risk of a traffic crash than the antidepressant treatment itself [34,35]. Furthermore, by improving clinical symptoms, including drowsiness, antidepressants can increase driving safety in these patients [35].

Regardless of the need for more research, health professionals should be aware of the possible risks and have to inform their patients individually on adverse effects of antidepressants and concomitant use with other medications [10].

Lastly, this study has some limitations. First, antidepressants dispensation cannot be considered as an equivalent to antidepressants consumption, as beneficiaries of our health system could have not consumed such medicines. Second, consumption in hospitals and by private prescriptions were not taken into account. However, in this case, it can be assumed that the biases produced when using dispensing data is not relevant, as antidepressants are financed medicines and their use require mandatory medical prescription. On the other hand, CONCYLIA does not include driver’s license information, so weighting was made on the basis of age and gender with the driver’s license census information, as performed in previous studies by our team [24,25,26,27].

## 5. Conclusions

In conclusion, the use of antidepressants was frequent especially with chronic and daily use. Between 2015 and 2018, the use of these medicines increased, especially the new N06AX antidepressants. In addition, concomitant use of antidepressants with other DIMs was relevant, which involved benzodiazepines. Importantly, SSRIs were consumed by a considerable proportion of the population, even if risk of using these medicines may be greater into the firsts days of use (these medicines belong to the DRUID category I, i.e., minor influence on fitness to drive).

Lastly, it is considered necessary to carry out new actions to provide more accurate information to healthcare professionals, patients, and drivers. It would be appropriate to improve prescription and dispensing systems for medications including alerts related to antidepressants. It is necessary to propose actions to avoid the unsafe driving of vehicles by patients under treatment with antidepressants. These dissuasive actions could be the inclusion of antidepressants into the battery for road detection tests, adapt traffic regulations to the use of these medicines, information campaigns, etc. [40].

## Figures and Tables

**Figure 1 pharmaceuticals-13-00061-f001:**
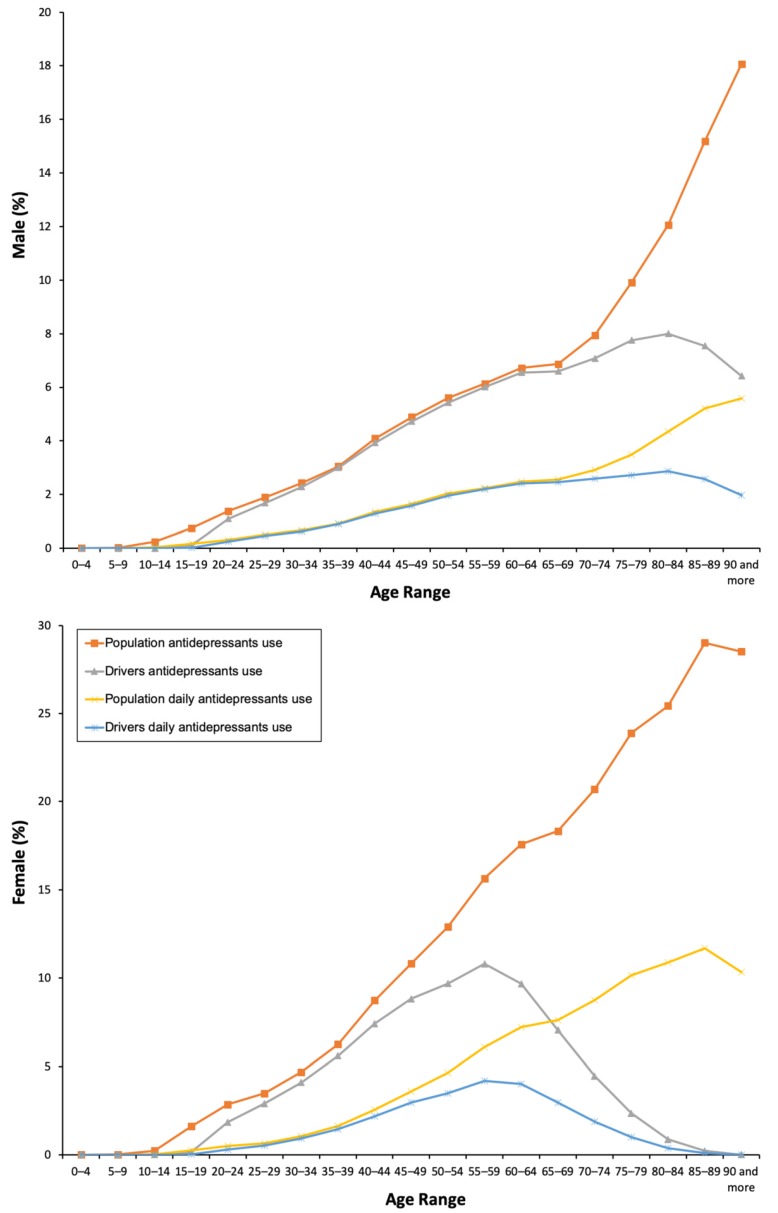
Frequency of antidepressants use by the general population and the driver population.

**Figure 2 pharmaceuticals-13-00061-f002:**
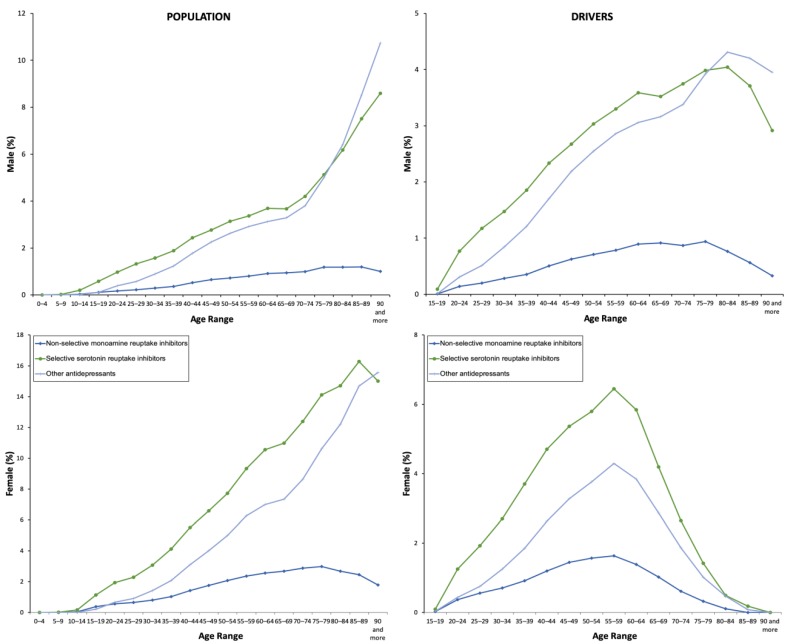
Frequency of use by the type of antidepressant.

**Figure 3 pharmaceuticals-13-00061-f003:**
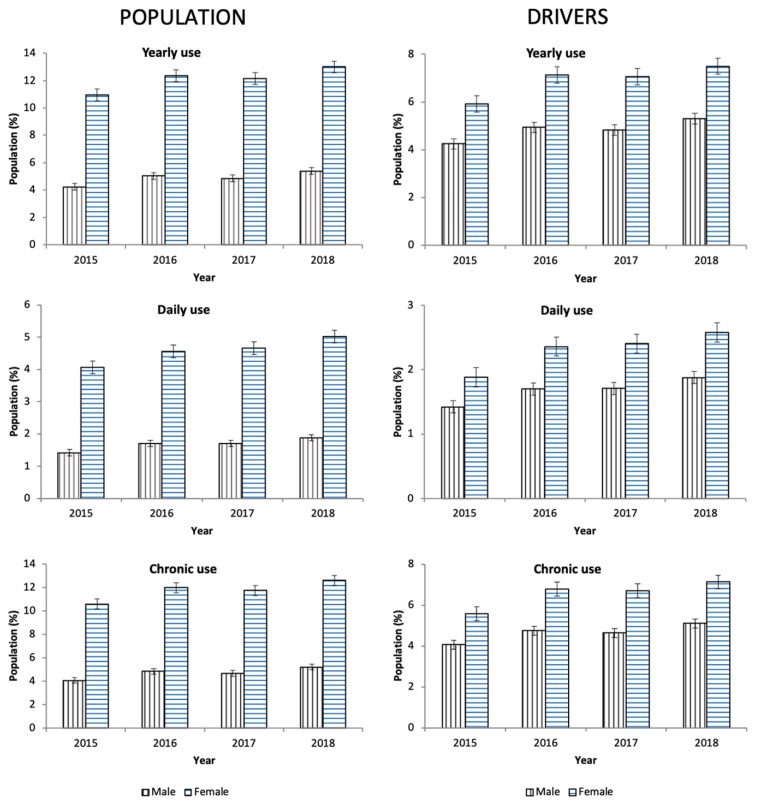
Evolution of antidepressants use in Castile and León (2015–2018).

**Figure 4 pharmaceuticals-13-00061-f004:**
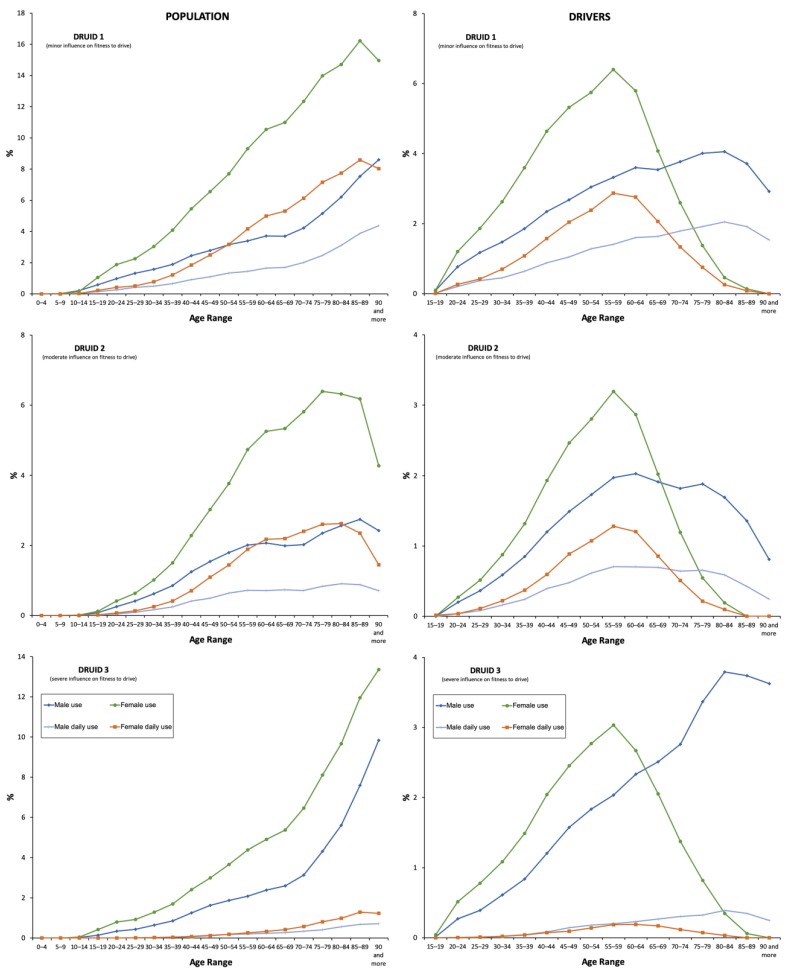
Frequency of use by Driving Under the Influence of Drugs, alcohol, and medicines (DRUID) classification.

**Table 1 pharmaceuticals-13-00061-t001:** Antidepressants consumption according to the CONCYLIA database and the Castile and León drivers’ license census data.

Type of Antidepressant	Population Using Antidepressants % (95CI)					Drivers Using Antidepressants % (95CI)				
	Yearly Use	Daily Use	Type of Use			Yearly Use	Daily Use	Type of Use		
			Acute	Subacute	Chronic			Acute	Subacute	Chronic
***Total Antidepressants***										
TOTAL	8.56 (8.52–8.59)	3.15 (3.13–3.17)	0.05 (0.04–0.06)	0.23 (0.22–0.24)	8.28 (8.25–8.32)	5.66 (5.62–5.70)	1.93 (1.91–1.95)	0.04 (0.03–0.05)	0.21 (0.20–0.21)	5.41 (5.37–5.44)
Male	4.87 (4.83–4.91)	1.68 (1.65–1.70)	0.03 (0.02–0.04)	0.15 (0.14–0.15)	4.70 (4.66–4.74)	4.83 (4.78–4.87)	1.68 (1.65–1.70)	0.03 (0.02–0.04)	0.16 (0.15–0.16)	4.64 (4.60–4.69)
Female	12.12 (12.06–12.17)	4.57 (4.54–4.61)	0.07 (0.06–0.08)	0.31 (0.30–0.32)	11.74 (11.68–11.80)	6.90 (6.83–6.96)	2.31 (2.27–2.35)	0.07 (0.06–0.08)	0.16 (0.15–0.17)	4.64 (4.59–4.70)
	Χ² = 1325.124; p = 0.001	Χ² = 4531.330; p = 0.001	Χ² = 32.612; p = 0.013	Χ² = 76.08; p = 0.001	Χ² = 7024.208; p=0.001	Χ² = 14360.161; p = 0.001	Χ² = 14615.499; p = 0.001	Χ² = 1160.409; p = 0.001	Χ² = 6083.759; p = 0.001	Χ² = 43897.32; p = 0.001
***Non-selective monoamine reuptake inhibitors***										
TOTAL	1.11 (1.10–1.13)	0.05 (0.04–0.05)	0.17 (0.16–0.17)	0.30 (0.29–0.31)	0.64 (0.63–0.65)	0.80 (0.70–0.90)	0.04 (0.03–0.05)	0.14 (0.13–0.15)	0.24 (0.24–0.25)	0.42 (0.41–0.43)
Male	0.58 (0.57–0.59)	0.03 (0.03–0.04)	0.08 (0.08–0.09)	0.16 (0.15–0.17)	0.34 (0.32–0.35)	0.61 (0.59–0.63)	0.04 (0.03–0.05)	0.09 (0.08–0.10)	0.17 (0.16–0.18)	0.35 (0.34–0.36)
Female	1.63 (1.61–1.65)	0.06 (0.05–0.06)	0.25 (0.24–0.26)	0.44 (0.43–0.45)	0.94 (0.92–0.96)	1.10 (0.99–1.12)	0.03 (0.02–0.04)	0.21 (0.2–0.22)	0.35 (0.34–0.37)	0.53 (0.51–0.55)
	Χ² = 48.521; p = 0.001	Χ² = 306.885; p = 0.001	Χ² = 36.373; p = 0.004	Χ² = 51.54; p = 0.001	Χ² = 392.684; p = 0.001	Χ² = 1837.676; p = 0.001	Χ² = 113.888; p = 0.023	Χ² = 1149.338; p = 0.013	Χ² = 2029.683; p = 0.001	Χ² = 3326.979; p = 0.001
***Selective serotonin reuptake inhibitors***										
TOTAL	4.99 (4.96–5.02)	2.22 (2.20–2.24)	-	0.23 (0.22–0.24)	4.76 (4.73–4.79)	3.32 (3.29–3.35)	1.33 (1.31–1.35)	-	0.19 (0.18–0.20)	3.13 (3.10–3.16)
Male	2.69 (2.66–2.72)	1.16 (1.14–1.18)	-	0.30 (0.29–0.31)	0.64 (0.63–0.66)	2.69 (2.65–2.72)	1.14 (1.12–1.16)	-	0.14 (0.13–0.15)	2.55 (2.51–2.58)
Female	7.22 (7.17–7.26)	3.24 (3.21–3.27)	-	0.32 (0.31–0.33)	6.90 (6.85–6.94)	4.26 (4.21–4.32)	1.62 (1.59–1.65)	-	0.26 (0.25–0.27)	4.00 (3.95–4.05)
	Χ² = 1009.602; p = 0.001	Χ² = 2947.450; p = 0.001	-	Χ² = 114.15; p = 0.001	Χ² = 4329.346; p = 0.001	Χ² = 6700.641; p = 0.001	Χ² = 10488.675; p = 0.001	-	Χ² = 1231.407; p = 0.001	Χ² = 23530.089; p = 0.001
***Other antidepressants***										
TOTAL	3.71 (3.69–3.74)	0.97 (0.96–0.99)	-	0.43 (0.42–0.44)	3.28 (3.26–3.3)	2.36 (2.34–2.39)	0.62 (0.61–0.63)	-	0.31 (0.30–0.32)	2.05 (2.03–2.07)
Male	2.31 (2.28–2.34)	0.54 (0.53–0.55)	-	0.29 (0.28–0.30)	2.02 (1.99–2.04)	2.24 (2.21–2.27)	0.56 (0.54–0.57)	-	0.29 (0.28–0.30)	1.95 (1.93–1.98)
Female	5.07 (5.03–5.11)	1.39 (1.37–1.41)	-	0.56 (0.55–0.57)	4.51 (4.47–4.54)	2.54 (2.50–2.58)	0.71 (0.69–0.73)	-	0.35 (0.33–0.36)	2.19 (2.15–2.23)
	Χ² = 661.836; p = 0.001	Χ² = 1423.581; p = 0.001	-	Χ² = 98.22; p = 0.001	Χ² = 3157.1; p = 0.001	Χ² = 5506.160; p = 0.001	Χ² = 4202.911; p = 0.001	-	Χ² = 2613.557; p = 0.001	Χ² = 16278.972; p = 0.001

Abbreviations: 95CI. confidence interval.

**Table 2 pharmaceuticals-13-00061-t002:** Other driving-impairing medicines (DIMs) used concomitantly with antidepressants.

ATC Code	Description	Population % (95CI)				Drivers % (95CI)			
		Yearly Use		Daily Use		Yearly Use		Daily Use	
		Male	Female	Male	Female	Male	Female	Male	Female
N05B	Ansyolitics	53.65 (53.36–53.94)	60.80 (60.51–61.09)	61.21 (60.52–61.89)	69.3 (68.91–69.68)	55.22 (54.75–55.7)	61.89 (61.42–62.37)	62.75 (61.97–63.53)	71.85 (71.09–72.60)
		Χ² = 1026.420; p = 0.001		Χ² = 845.748; p = 0.001		Χ² = 6479.230; p = 0.001		Χ² = 2593.569; p = 0.001	
N02A	Opioids	20.84 (20.5–21.17)	28.61 (28.38–28.84)	20.77 (20.20–21.34)	31.76 (31.37–32.15)	20.53 (20.14–20.91)	23.33 (22.92–23.74)	20.43 (19.78–21.08)	26.49 (25.75–27.23)
		Χ² = 399.964, p = 0.001		Χ² = 378.297, p = 0.001		Χ² = 3344.400, p = 0.001		Χ² = 1178.273, p = 0.001	
N02B	Other analgesics and antipiretics	16.15 (15.84–16.45)	20.59 (20.39–20.8)	15.96 (15.45–16.47)	22.53 (22.18–22.88)	15.40 (15.06–15.74)	16 (15.64–16.35)	15.08 (14.50–15.66)	17.67 (17.03–18.31)
		Χ² = 229.200, p = 0.001		Χ² = 201.621, p = 0.001		Χ² = 2612.735, p = 0.001		Χ² = 939.163, p = 0.001	
N05C	Hypnotics and Sedatives	17.33 (17.01–17.64)	17.16 (16.97–17.36)	18.63 (18.08–19.18)	19.81 (19.48–20.14)	16.97 (16.61–17.33)	14.64 (14.30–14.98)	18.56 (17.93–19.19)	18.85 (18.19–19.50)
		Χ² = 235.223, p = 0.001		Χ² = 188.963, p = 0.001		Χ² = 2087.480, p = 0.001		Χ² = 835.451, p = 0.001	
N03A	Antiepileptics	19.14 (18.82–19.47)	16.28 (16.09–16.47)	24.48 (23.88–25.08)	20.93 (20.59–21.27)	20.24 (19.86–20.62)	17.67 (17.30–18.04)	26.30 (25.59–27.01)	25.58 (24.84–26.31)
		Χ² = 362.380, p = 0.001		Χ² = 359.500, p = 0.001		Χ² = 1878.187, p = 0.001		Χ² = 775.130, p = 0.001	
N05A	Antipsychotics	20.73 (20.40–21.06)	15.43 (15.25–15.62)	29.39 (28.75–30.03)	20.20 (19.86–20.53)	20.71 (20.32–21.09)	13.44 (13.11–13.77)	30.76 (30.02–31.51)	20.96 (20.27–21.64)
		Χ² = 1082.860, p = 0.001		Χ² = 876.871, p = 0.001		Χ² = 1435.496, p = 0.001		Χ² = 610.888, p = 0.001	
**Average of Driving Impairing Medicines. Population Antidepressants Use**		2.74 ± 1.14	2.76 ± 1.18	2.86 ± 1.12	2.92 ± 1.23	2.73 ± 1.31	2.59 ± 1.37	2.89 ± 1.35	2.88 ± 1.43
		t = −2.951; p = 0.003		t = −7.255; p = 0.001		t = 20.608; p = 0.001		t = 1.150; p = 0.250	

Abbreviations: 95CI. confidence interval.

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
