# Peer review of "Trends in Antidepressants Use in Spain between 2015 and 2018: Analyses from a Population-Based Registry Study with Reference to Driving"

_pharmaceuticals, 2020, doi:10.3390/ph13040061_

Round 1

Reviewer 1 Report

The study is on trends in antidepressant use in Spain from 2015 to 2018. It is a timeous and relevant paper to public health experts and the public alike. However, I think the paper may require some additional changes to improve clarity and substance to the readers. Here are a few comments:

Major

1. The authors state this paper is on trends in antidepressant use but there is no statistical analysis or approach that actually show anything about the trends. The authors use Chi-square tests and Student's t-tests to compare the prevalence and mean number of antidepressants in males and females and by DIM types. But these data combined 2015 to 2018 data which means we can not infer anything about trends. Although authors show graphs of yearly consumption, they should conduct a formal trend test such as the Cochran Armitage test to show if there was any change in antidepressant use from 2015 to 2018 otherwise the title should be changed.

2. Figure 3 presents R2 values which I assume are meant to  show the correlation across the years. It's inappropriate to present the R2. Conduct a trend test . See comment above.

3. There is repetitive information in the introduction and methods e.g. line 84-86 and line 99-100

Minor

There are a lot of grammatical and stylistic errors throughout the paper. For instance rewrite sentence 84-86, replace the word 'Interestingly' with 'However' in line 83, replace 'y'  with 'and' in line 87.

Reviewer 2 Report

This is a well written article that describes the increase in antidepressant prescriptions, and presumably use, in a region of Spain.  It provides useful data on these trends, however I advise caution on inferring use of these drugs and risk of collisions.  There is overwhelming evidence that treating people with symptoms of significant depression is far better than not, and while the older TCAs did have some initial adverse effects on driving before patients developed tolerance and their moods increased there is little credible evidence that the SSRIs/SNRIs have an elevated crash risk above any risks associated with any residual depression.  This issue needs to better discussed to avoid the view that these drugs should not be given or that people on these drugs are warned not to drive.  The various European and ICADT do need harmonization and need to separate the disease from the treatment.  In the papers cited such as by Brunnauer (2 reviews) and Ravera the reviews largely focus on experimental studies not crash risk studies that look at the overall crash risk and can take use of other medications into account (which is very common) – a recent paper from Australia reinforces their previous study published in 2014 that shows no increase in crash risk for antidepressants (Drummer 2019 Acc Anal Prev).

Citations from Brunnauer A (2013) and (2017) it says “There is evidence that the SSRIs (citalopram, escitalopram, fluoxetine, fluvoxamine, sertraline, paroxetine) and the SNRI venlafaxine have no deleterious effects on driving ability”.  This conclusion is similar in their 2017 review; and in Ravera (2012) it concludes that “Regarding the selected experimental studies, it could be argued that experimental data suggest that SSRIs do not constitute a high risk to traffic safety unless used at high dosages or combined with other psychotropic substances” are not reflected in the submitted manuscript.

Round 2

Reviewer 1 Report

Authors have addressed major concerns. I would suggest authors proofread for typos and better sentence structure before final publication.